# Rapid, without focus stacking, 3D photogrammetric digitization of cockroaches

**Juliette Berger**[ID][☉][*], **Barbara Bignon**[☉], **Raphaël Cornette**, **Frédéric Legendre**[ID][‡*], **Arnaud Delapré**[ID][‡*]

Institut de Systématique, Évolution, Biodiversité (ISYEB), UMR 7205, Muséum national d'Histoire naturelle, CNRS, Sorbonne Université, EPHE-PSL, Université des Antilles, Paris, France

☉ These authors contributed equally to this work.
‡ These authors jointly supervised this work.
* juliette.berger@sorbonne-universite.fr (JB); frederic.legendre@mnhn.fr (FL); arnaud.delapre@mnhn.fr (AD)

## Abstract

Natural history collections are seen as treasure troves we need to both preserve and study. Campaigns of 2D and 3D digitization have emerged in numerous institutions as an opportunity to maximize specimen's diffusion while limiting the risk associated to their manipulation. 2D and especially 3D models can be used for various scientific purposes. Because of different obstacles (time, technical limitations, cost, etc.), the digitization of small and numerous objects, like insect specimens, remains to be improved. Among the existing options, photogrammetry is generally less expensive than µCT-scan, two of the main methods for digitizing objects, but it remains time-consuming for small objects because focus staking—which involves a multiplication of shots—is strongly recommended to increase the depth of field. Here, we present a fast and inexpensive photogrammetric pipeline that generates 3D models of cockroaches of sufficient quality for morphometric geometric analyses. By focusing on a region of interest in the specimens—identified according to the goal of the digitization—the depth of field is reduced by comparison with the one encompassing the whole specimen. Thus, we eliminated the need for focus stacking. We produced 3D models for 62 species and compared 13 of the photogrammetric 3D models qualitatively and quantitatively with those obtained from µCT-scans of the same 13 species. We conclude that the 3D models produced with our pipeline are of sufficient quality to perform geometric morphometric analyses, which will be published elsewhere in a companion paper. Despite a few limitations, we hope that our pipeline will generate opportunities for the study of small objects like insects, one of the most species-rich group on Earth and in natural history collections.

**Data availability statement:** All files are available from the FigShare repository at https://figshare.com/s/f4120b6b4bcebcef038e.

**Funding:** This work was supported by French National Research Agency grant no ANR-19-CE02-0023 (project Sociogenomics) to Frédéric Legendre. This work was also supported by the MRI platform, member of the national infrastructure France-BioImaging supported by the French National Research Agency (ANR-10-INBS-04, «Investments for the future»), the labex CEMEB (ANR-10-LABX-0004) and NUMEV (ANR-10-LABX-0020).

**Competing interests:** The authors have declared that no competing interests exist.

## Introduction

The digitization of natural history collections (NHC) is of major interest for the preservation of specimens and the dissemination of scientific knowledge [1–3]. NHC provide a vast record of the Earth's biodiversity, which has been and continues to be used to address fundamental questions in ecology, systematics, or evolution [2,4]. Despite the current biodiversity crisis imposing practical and ethical restrictions, the importance of NHC cannot be overstated. Their importance has even been revitalized by, for example, the development of new tools and methods in sequencing or bioinformatics [5–7]. Among NHC, insects stand out for their diversity and wealth of species and specimens [5]. Insects are also of relatively small size, a well-known challenge for 3D digitization [8]. By providing three-dimensional (3D) models of insect specimens, digitization helps preserve fragile specimens by limiting unnecessary handling, while making them available to those who cannot physically access them [9,10]. These models can be used for morpho-anatomical observations and analyses so that digitization contributes not only to sharing biodiversity data but also to entomological research and technological advances in 3D imaging and analysis [11,12].

Several techniques allow digitizing insect specimens, the main ones being X-ray micro-computed tomography (µCT-scan) and photogrammetry (also known as structure from motion photogrammetry) [13,14]. Aside from manipulation risks, both are relatively non-invasive, which is crucial to ensuring the durability and usefulness of NHC. However, both have their strengths and limitations. Photogrammetry relies on external images to create realistic 3D models of specimens, true to both forms and colors [13–17]. It is a relatively non-expensive method [18], valuable not only for visualization purposes, but also for digital preservation of specimens in collections—which can be revisited over time— or for precise and non-invasive morphometric analyses [10,13,19]. It is, however, useless when internal structures need to be examined [14]. X-ray tomography, on the other hand, enables in-depth exploration of internal structures, including details of internal organs and soft parts [20]. But µCT-scans require a dedicated infrastructure, a lot of storage space and computing resources. In addition, pinned specimens may require special preparation or conditions to avoid artifacts caused by the pin: while it is best to remove the pin—which can be a delicate and thorny undertaking with old and fragile type specimens—it is also possible to adjust the intensity of the X-rays or use special algorithms after acquisition. Choosing between these techniques depends on the specific purpose of the digitization and the trade-offs between cost, resolution, and equipment availability [13,21].

As cost is a major constraint in academia, photogrammetry is a widespread solution, especially as many steps can be automated using specialized software [22,23]. It offers an efficient, non-invasive solution for obtaining accurate information about the shape and structure of objects from photographs [24–26]. Briefly, photogrammetry can be delineated in five steps [24,26,27]: 1) *Image acquisition*: Photographs of the object are taken from different angles. This is the only 'physical' step, the subsequent steps being carried out using dedicated software for the model reconstruction. 2) *Identification of common points*: This means identifying the positions of the camera

relative to the position of the specimen. It can be achieved by using known landmarks (control points) or common features automatically detected in the images. 3) *3D point extraction*: software uses the parallax information in the images to calculate the three-dimensional position of the object's landmarks. These points are then used to create a 3D point cloud representing the geometry of the object. 4) *3D Modeling*: From the point cloud or the depth-maps, a three-dimensional surface is reconstructed to form the 3D model of the object. 5) *Texturing*: The original images are projected onto the surface of the 3D model, giving it a realistic texture. The resulting model is a mesh of dimension faces, or mesh surface. The accuracy of the results depends primarily on the quality of the images and their overlap.

However, for small organisms such as insects, photogrammetric digitization often requires imposing setups and pipelines, creating many obstacles to rapid and inexpensive digitization [22,28]. The low depth of field of high-magnification lenses is a major constraint, which is overcome by focus stacking to acquire an extended depth of field (EDOF) [27,29–32]. But this approach comes at the cost of rapidity as focus stacking requires multiple pictures to be taken at different focal points, which significantly increases acquisition time. Here, we present a comparatively rapid and cost-efficient photogrammetric approach for insects. By focusing on a single part of the body (i.e., the region of interest; in this case, cockroach pronotum), the depth of field that needs to be covered to obtain a sharp image is reduced by comparison with the one encompassing the whole specimen. This reduction eliminates the need for focus stacking while still capturing the rest of the insect, albeit at a lower quality. Importantly, this pipeline can produce high-quality 3D models of the structure of interest that realistically capture the complex shape of these small organisms and, because of its relative rapidity, models that can be used for comparative analyses on large samples. To validate this pipeline, we compared qualitatively and quantitatively the 3D models obtained by our approach with those obtained by µCT-scans for a few specimens.

## Materials and methods

### Biological context and region of interest

Our final goal, as reported in a ecomorphological study [33], is to investigate morphological adaptations to fossorial habits in cockroaches. We focused on the pronotum, the first dorsal thoracic segment, because the pronotum is of systematic importance, is a functionally versatile structure in cockroaches, and can have a complex shape (Fig 1). The pronotum is notably used in some species to dig in the ground or to make galleries in wood [34,35]. Studying the pronotum of cockroaches in 3D, an unprecedented undertaking, allows a richer and more precise characterization of its shape than is usually done in systematics [36–38].

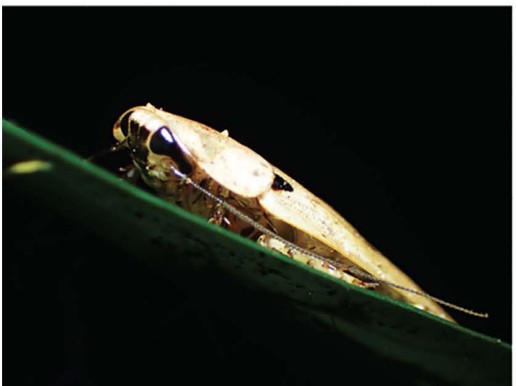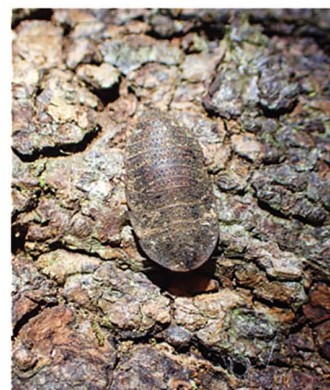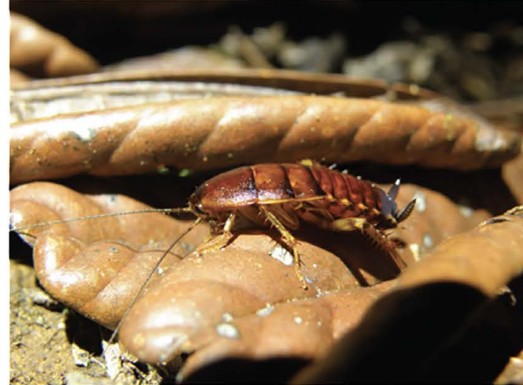

**Fig 1. The pronotum, the first dorsal thoracic segment, is a functionally versatile structure in cockroaches.** In numerous species, it covers most of, if not all, the head. From left to right: *Rhabdoblatta* sp. and *Laxta* sp. (Australia), *Angustonicus* sp. (New Caledonia). Credits: Frédéric Legendre.

We selected 62 species of cockroaches from the collections of the Muséum national d'Histoire naturelle (MNHN, Paris), sampling the three known cockroach superfamilies: Blaberoidea, Blattoidea and Corydioidea (Table 1). Species were selected according to their habitats and availability in the MNHN collection [33]. The size of the specimens ranged from 5 to 45 mm, with four species smaller than 10 mm. Sixteen species had more or less distinct and numerous bristles, especially around the pronotum, while 26 species were reflective, 15 of which were black. The presence of bristles, reflection and a black coloration constitute three challenging conditions for obtaining 3D models by photogrammetry [13,18,21,25,27,31,39].

Microtomographic acquisitions were obtained for 13 of the 62 species, for comparison purposes, using the EasyTom 150 microtomograph (X ray source between 40–150 kV, power up to 75 W, resolution up to 5 µm) at the platform Montpellier, Ressources, Imagerie (MRI). The size of the specimens varied from 5 to 15 mm, and the settings varied accordingly. The focal spot diameter ranged from 8 to 20 µm, the power from 10 to 30 W, and the voxel size ranged from 6 to 19 µm. 3D models were reconstructed using the X-Act software (RX Solution, Chavanod, France), configured with 720 projections over 360°. Some of the photogrammetry-challenging specimens (black, reflective or with bristles) were among these 13 species (Table 1).

## Photogrammetry set-up installation

All pictures were taken with a Nikon D7500 DSLR (Digital Single Lens Reflex) camera equipped with an AF-S DX Micro NIKKOR 40 mm f/2.8G lens, all mounted on a tripod (Fig 2). Depending on the size of the specimen, one or more Kenko extension tubes (12, 20, 36 mm) were added to the lens to increase magnification. We used a fixed focal length and manual focus. A Hoya circular polarizing filter was attached to the lens to reduce reflections on the specimens. The photographs were taken in a Godox LST40 mini photo studio (60 x 60 x 60 cm), equipped for lighting with three LED strips

**Table 1. Number of species sampled within each superfamily to generate photogrammetric 3D models (N = 62). The sizes of the specimens as well as their familial and subfamilial attributions are provided. When three species or more were sampled in a subfamily, only the smallest (min) and largest (max) sizes are reported. Each * indicates a microtomographic acquisition for one species (N = 13).**

| Superfamily | Family | Subfamily or tribe | Number of species | Size (mm) (min – max) |
|---|---|---|---|---|
| **Blaberoidea** | Blaberidae | *incertae sedis* | 1 | 24.5 |
| | | Blaberinae | 3 | 34.9 - 43 |
| | | Diplopterinae | 1 | 20.5 |
| | | Epilamprinae | 2 | 19.4–28.8 |
| | | Gyninae | 3* | 12.7 - 27 |
| | | Panchlorinae | 1 | 20.9 |
| | | Panesthiinae | 7** | 23.3–36.6 |
| | | Perisphaerinae | 3* | 14.1–27.3 |
| | | Zetoborinae | 7* | 20–38.2 |
| | Blattellidae | Parcoblattini | 1 | 14.3 |
| | Nyctiboridae | – | 1 | 15.3 |
| | Pseudophyllodromiidae | Pseudophyllodromiinae | 1 | 9.3 |
| **Corydioidea** | Corydiidae | Corydiinae | 20****** | 15.3–42.6 |
| | | Euthyrrhaphinae | 3 | 5.2–9 |
| **Blattoidea** | Blattidae | Polyzosteriinae | 2 | 16.4–25.7 |
| | Cryptocercidae | Cryptocercinae | 3** | 19.7–25.5 |
| | Lamproblattidae | Lamproblattinae | 1 | 23.7 |
| | Tryonicidae | Tryonicinae | 2 | 10.9–16.4 |

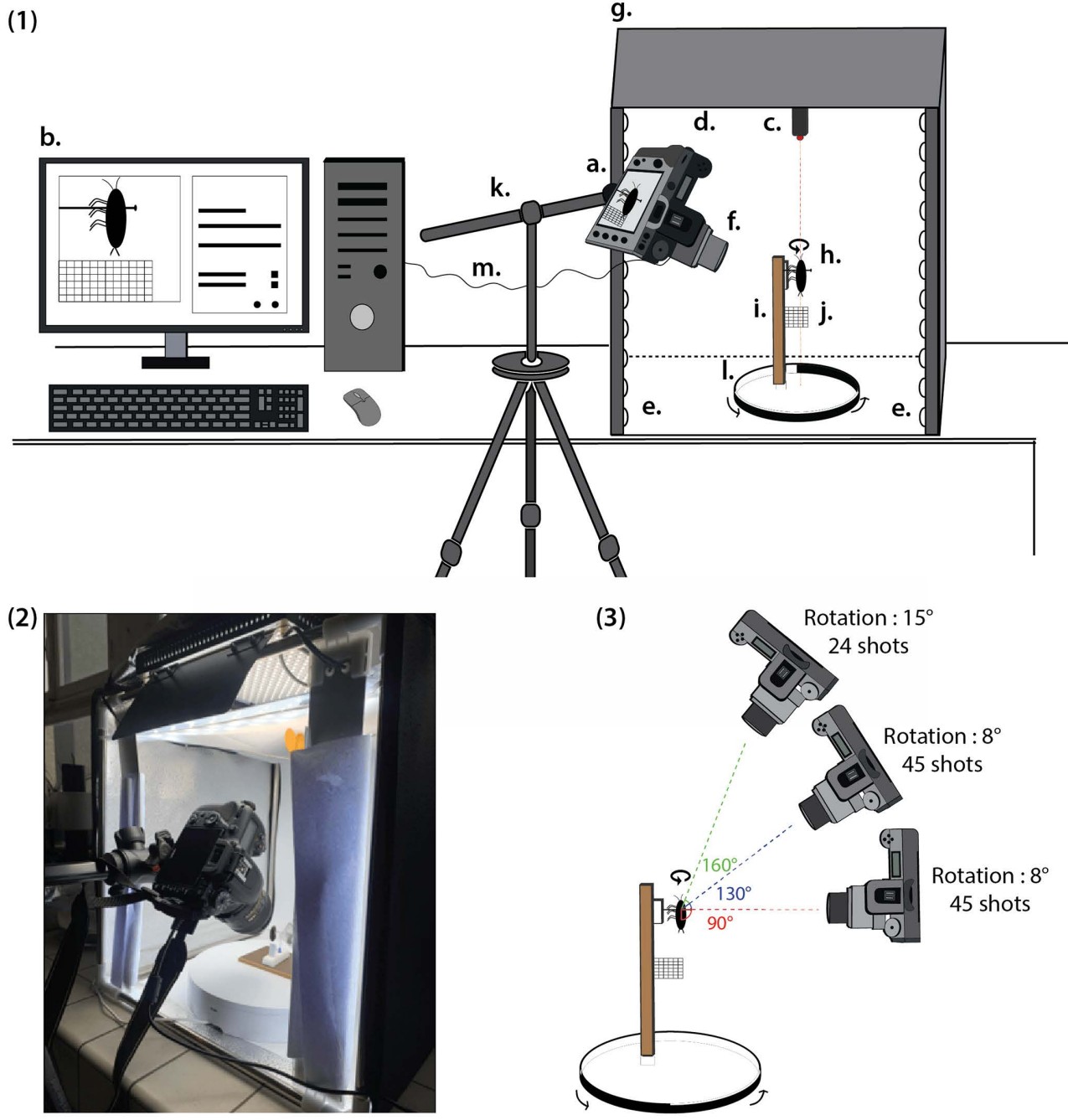

**Fig 2. Experimental setup and photogrammetric acquisition principle.** (1) Set-up installation with the camera (a) mounted on a tripod (k) and connected to a computer and its dedicated software (b), and the specimen (h) placed on a sample holder (i) inside the mini-photo studio (g) and above the center of a turntable (l) controlled by Bluetooth; c) laser, d) LED panel, e) LED strip, f) lens, j) scale, m) USB cable; (2) Photograph of the mini-studio installation (3) Schematic diagram illustrating photogrammetric acquisition using three shooting angles (90°, 130° and 160°) between the vertical axis of the sample and the camera.

with a color temperature of 5600 K, adjustable in intensity, and a Dörr DLP1000 LED panel, adjustable in temperature and intensity [13]. These LEDs provided a uniform lighting and, to reduce cast shadows, all the lamps were covered with white paper or fabric. The whole set-up is portable.

Specimens were mounted on holders—we designed and produced three different sizes of holders to accommodate differences in specimen size—and placed on a Foldio 360 turntable, which was controlled and connected to a smartphone via Bluetooth. A millimeter scale was added to the sample holder. Once mounted, the exact position of the specimen is adjusted to its center of rotation, allowing an accurate 360° view of the entire specimen. The rotation axis of the specimen is identified by a laser pointer, positioned vertically (at 90°) from the center of the turntable. The digital camera was set up in portrait mode because of the elongated form of these insects—and thus maximizing the number of pixels representing the specimens—and connected via USB cable to Nikon Camera Control Pro 2 version 4.0.11 (Nikon Corporation).

The cost of the full set-up is itemized in Table 2. It amounts to *ca.* 3.5 k$ and can be reduced by using free software [17,40].

**Step 1: Image acquisition process.** Once the insect position has been adjusted to its center of rotation, image acquisition can begin. The camera was focused on the pronotum of cockroaches, which is the region of interest. Before each series of pictures, the mounting and center of rotation of the specimen were checked, as well as the reflection of light on each specimen. The optimum aperture, shutter speed, white balance and sensitivity settings were determined using the digital camera's options and refined using Nikon Camera Control software: usually an aperture of f22 to allow a wide depth of field, a shutter speed comprised between 1/10s and 1/2s to compensate for aperture, and a sensitivity of 64–100 iso for a good definition and noise reduction. Mirror up mode (Mup) was also activated to reduce blur. The exposure was manually adjusted according to the morphological features of the specimen.

To optimize the alignment process during reconstruction and guarantee a detailed 3D model, good overlap between images is required. After a few tests with two to four series of photographs at different angles, three series were found to be necessary and sufficient for accurate modelling in our case [see also 41]. The choice of angles depends on the size of the specimen, the distance at which the image is taken, the digitization equipment used, and the region of interest. Here, three series of photographs were taken at three different angles between the vertical axis of the cockroach and the lens: 90°, 130° and 160° (Fig 3). Each series took photographs of the specimen over a full rotation (360°). During a full rotation, for the first two angles (90° and 130°), the turntable rotated 8 degrees between each shot; for the third angle (at 160° because we focus on the pronotum, in the upper part of the specimen), the turntable rotated 15 degrees between each shot—15 degrees were a good compromise here between quality and time; it has also been identified elsewhere as the

**Table 2. The detailed cost of our 3D insect modeling system.**

| Device | Price in USD |
| --- | --- |
| Nikon D7500 DSLR | 1250 |
| AF-S DX Micro NIKKOR 40 mm f/2.8G lens | 360 |
| 3 Kenko extension tubes (12, 20, 36 mm) | 200 |
| Manfrotto tripod | 320 |
| Hoya circular polarizing filter | 50 |
| Godox LST40 mini photo studio | 140 |
| Dörr DLP1000 LED panel | 270 |
| Foldio 360 turntable | 160 |
| Nikon Camera Control Pro 2 | 180 |
| Agisoft Metashape Professional | 650 |
| **Total** | **3580** |

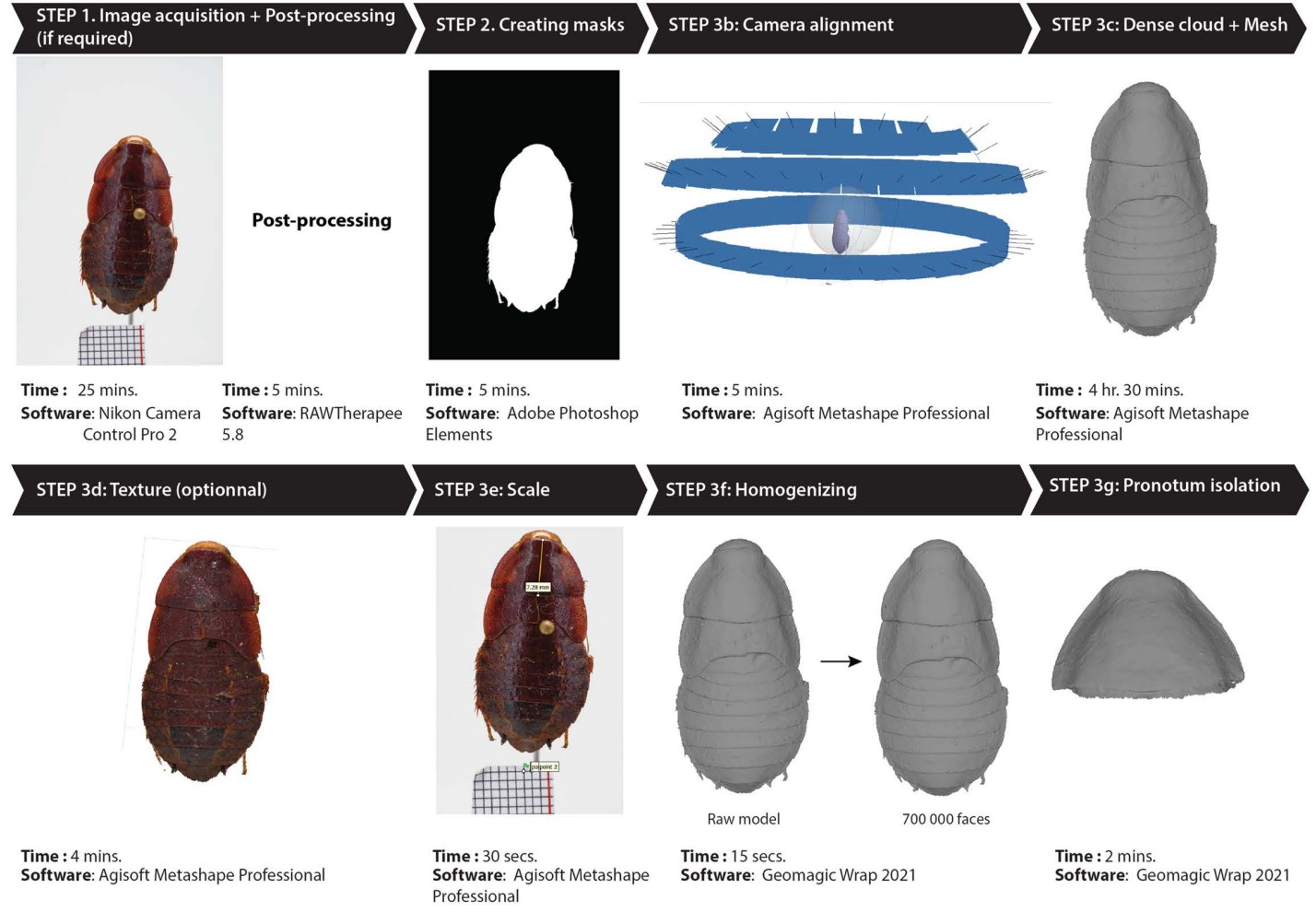

**Fig 3. Workflow overview using a specimen of the species *Galiblatta cribrosa* Hebard, 1926 with the software used and an estimation of the time for each step.** Time estimations were averaged from the metadata of the files generated at each step. 1. Detail of the physical setup shown in Fig 2; 2. Creation of a mask on a post-processed JPEG photo; 3b. Alignment of the camera, represented by blue rectangles for the three viewing angles; 3c. 3D mesh from dense point cloud; 3d. Model texture generation from photo color; 3e. Scale for subsequent measurements; 3f. Homogenization of the number of faces in the dataset (reduced to 700,000 faces); 3g. Isolation of the region of interest for future analysis (here, the pronotum).

optimal lowest value [41]. It resulted in a total of 114 pictures taken over a period of 25 minutes. Pictures were saved in Nikon's raw (.NEF) and compressed JPEG Fine formats. Photographs in JPEG format were used to control the images obtained during image acquisition, while those in RAW format were used for post-processing. To maximize reconstruction quality, a 90° photograph of the specimen in dorsal view (RAW format) was selected and, in post-production, we proceeded with sharpness, brightness adjustment and fill-in light using RAWTherapee 5.8 free software (Horváth and Raw-Therapee Development Team). The retouched profile was then applied to the full set of photos of the same specimen. All photographs in RAW format were exported in JPEG format for 3D model reconstruction.

**Step 2: Creating masks.** The integration of masks in 3D photogrammetric reconstruction provides several significant advantages [22,41,42]. The purpose of a mask is to hide the background and isolate the subject by accurately delimiting the areas of interest in each image, thereby excluding any unwanted elements and enhancing reconstruction accuracy. Masks also simplify the *identification of common points* step because unwanted elements are excluded, thus optimizing

the photo alignment process. Overall, using masks in 3D photogrammetric reconstruction reduces memory and computing power requirements, speeds up the reconstruction process, and improves accuracy by eliminating superfluous information.

To create a mask for a given specimen, we imported five photographs from different angles and with the highest sharpness into Adobe Photoshop Element 2021 (Adobe Systems, USA). On each photo, the specimen was selected with high precision using the "magnetic lasso" tool. A fusion mask was then created from this selection and exported in binary mode (black & white) in JPEG format. Finally, the mask files were imported into Metashape using the mask import function.

**Step 3: 3D reconstruction and isolation of the region of interest.** Although free alternatives exist [e.g., 17,43,44], 3D reconstructions were performed with Agisoft Metashape Professional version 2.0, largely used by the scientific community [e.g., 17,45], following the workflow of [28]:

Step 3a: For each sample, the set of photographs and corresponding masks are imported into the photogrammetric processing software Agisoft Metashape Professional. Note that there was no specific calibration step, as the camera and its focal were automatically recognized by Metashape.

Step 3b: In order to position and orientate the photographs in space and thus visualize an initial point cloud (sparse cloud or tie point cloud) representing the sample, an alignment is performed after importing and associating the masks with the dataset. The internal and external orientation (IO & EO) parameters of the images were calculated automatically in Metashape. Indeed, during the photo alignment procedure, Metashape estimates the interior and exterior camera orientation parameters, including nonlinear radial distortions. To successfully estimate the camera orientation parameters, the information on approximate focal length (pix) is required. This information is extracted automatically from the EXIF metadata.

The alignment is performed with a medium precision parameter, which speeds up the process (Fig 3). The model is then filtered to minimize reconstruction uncertainty and correct projection errors. Camera positions are optimized, exported and re-imported with the "highest" accuracy option. As the camera positions have already been determined, this method is much faster than aligning directly with a high accuracy setting. Once the alignment is complete, the calculated camera positions and a tie point cloud are displayed.

Step 3c: A dense point cloud is then generated, representing the 3D geometry of the sample. The quality and filter parameters are set to "Ultra High" and "Mild", respectively. The "Mild" setting is required for subsequent mesh reconstruction based on depth maps—each pixel contains a depth value representing the distance between the camera and the object surface. These depth maps are then transformed into partial point clouds for each image and merged into a single point cloud. In this way, Metashape can construct a mesh of numerous triangles connecting the points in the dense cloud. Using depth maps as a data source allows more efficient use of 2D image information and requires fewer resources compared to point cloud-based reconstruction. This mesh is created using the "Build mesh" option with the following parameters: strict volumetric masks enabled, quality set to "Ultra High", and limited to 1,000,000 faces.

Step 3d (optional): A model texture is constructed using the original photographs. This step creates a textured model that retains the visual details of the scene.

Step 3e: Photography, and by extension photogrammetry, does not provide information about the scale of the object or scene being digitized. But it is pivotal for geometric morphometric analyses. Thus, with the "Add marker" function, we manually defined the scale by positioning two control markers on four to six photographs of the series at 90°, which helps to minimize parallax problems. These markers are placed on the millimeter scale of the photograph. The distance between the two markers is entered in the interface and the scale information is updated. Now, the model integrates its true scale, which was validated by measuring specimens.

Step 3f: The 3D model is then exported in PLY format and decimated to 700,000 faces—this figure was previously tested and found to be a good compromise between mesh size and fidelity of the 3D model—using the 3D mesh

processing Geomagic Wrap 2021 software (Geomagic Company, United States). The purpose of this step is to standardize all the data and to speed up model landmarking by generating smaller files.

Step 3g: Focusing the camera specifically on the pronotum, we ensured that it was captured with optimum sharpness. We then isolated pronotums for downstream analyses. To do this, the pronotums were cut out using Geomagic Wrap, parts of the 3D model that do not belong to the pronotum being manually eliminated.

For all the 3D reconstructions, we used a computer with two processors at 3.8 GHz, 96 GB RAM and a Radeon Pro WX 5100 graphic card.

## Model landmarking

Although 3D models of whole specimens were generated, 3D geometric morphometric analyses [46] were conducted only in the pronotum, the region of interest. For a given pronotum, we used two 3D anatomical landmarks and 14 sliding semi-landmarks of curves to represent its contour [47], which was combined with a mesh of 311 sliding surface semi-landmarks to capture its shape [48]. First, we built a template to be used as a reference for setting landmarks on the 26 3D pronotums obtained (13 from each acquisition method). The template was based on the specimen of *Galiblatta cribrosa* Hebard, 1926, whose pronotum shape is the closest to the average shape of the samples at hand. The landmarks were always placed in the same order (as in the template) to preserve the correspondence landmarks from one specimen to the next. The template surface sliding semi-landmarks were then projected and slid onto the other models [49] using the R packages Morpho and rgl [50,51]. Note that a potential operator bias was previously estimated with a repeatability analysis (N = 5) performed on three species (*Angustonicus* sp. Grandcolas, 1997, *Hemelytroblatta cerverae* (Bolívar, 1886) and *Salganea raggei* Roth, 1979). It was found to be negligible compared to the biological variability.

## Quality of the 3D models: Quantitative comparison using geometric morphometrics

Of the 62 digitized specimens, 13 specimens were acquired using both photogrammetry and computed microtomography (µCt-scan). For these specimens, we performed geometric morphometric analyses on the region of interest to assess whether the method of acquisition (photogrammetry or µCt-scan) could affect the downstream analyses. In other words, we evaluated the quality of the 3D models generated by our photogrammetric pipeline through visual comparisons of model pairs and a quantitative approach. This allowed us to assess the potential of our insect photogrammetric models for interspecific studies of evolutionary morphology [24,52].

To quantify the differences between the two approaches, pairs of 3D models were aligned in MeshLab (version 2020.07) [53]. During this process, control points are defined on both models to be aligned, and an alignment algorithm uses these points to achieve precise alignment. MeshLab's 3D comparison function was then used to calculate the distances between the models, generating a deviation map. Subsequently, the "Distance from reference mesh" filter was used to calculate the distance per vertex between a target mesh and a reference mesh. This distance was stored in the quality of the vertices, which were then colored according to this distance.

To assess the (potential) bias due to the two methods of acquisition, we carried out a Principal Component Analysis (PCA) on the Procrustes coordinates obtained using the *gpagen* function in the geomorph package [54]. For each species, we therefore visualized the variability between the two modes of acquisition, as well as the interspecific variability. A MANOVA (multivariate analysis of variance; [55]) was finally used to assess the relationship between geometric shape and digitizing techniques.

## Results and discussion

We produced a set of 71 pronotum models in 3D (58 by photogrammetry and 13 by µCt-scan) for 62 species from the collections of the MNHN, and representing 9 of the 13 Blattodea families. A gallery of all the 3D models obtained by photogrammetry and µCt-scans are available at 10.6084/m9.figshare.27325605. This link also contains a spreadsheet file with several metrics including post-adjustment statistics associated to each model.

As already underlined, the photographic set-up is portable, inexpensive and provides a fast workflow [18]. The hardware installation requires relatively little physical space, and the full set-up—computer not included—costs around 3,500 USD (including 650 USD for software, whereas free alternatives do exist [17,40]; Table 2). It is also fast. The reconstruction of each 3D model in ultra-high quality with our computer configuration and Metashape took an average of 4 hours and 41 minutes (steps 3b-3d), and much of the software chain is automated (dense cloud, mesh and texture creation), requiring only minimal supervision. Model file sizes averaged 23 MB and ranged from 13.06 to 74.82 MB, which is worth considering when storage capacity and online sharing are limited [13]. The number of vertices in the reconstructed raw models averaged 1,289,763, ranging from 475,600–2,518,759.

## 3D entomology models without focus stacking for comparative morphology

Focus stacking is typically used in photogrammetry for small objects [21,31,39,56]. It consists of combining several images taken from the same position but with different focus in order to increase the depth of field. Its main drawback, however, is the increased acquisition time, as taking pictures with different focal points is time-consuming. In addition, focus stacking requires specialized equipment [56], as some automated photogrammetry systems may not be compatible with focus stacking. Furthermore, the process usually involves post-processing to merge the images, requiring additional time.

By opting for a photogrammetry method that excludes focus stacking, particularly suited to small and complex subjects, and focusing on a precise area, we achieve a satisfactory compromise between quality and convenience. The quality was assessed visually, as well as quantitatively by comparing 3D models produced by photogrammetry and µCt-scan for 13 species [24,52].

Visual inspection of the 3D models suggests that some specimens proved more challenging to reconstruct by photogrammetry than others (Fig 4). For instance, the 3D models of species with bristles on the edges of the pronotum (e.g., *Heterogamisca kruegeri* (Salfi, 1927), *Hemelytroblatta africana* Linnaeus, 1758) show some irregularities. The surface is granular, and the contours are not precise enough for landmarking and 3D morphometric analyses. The color of the pronotum can also be troublesome [13], as in *Phortioeca peruana* Saussure, 1862, whose pronotum is dark brown except on the anterior margin where it is light brown or even transparent. Its pronotal surface is in addition punctuated. This results in a 3D pronotum model with poorly defined edges and an irregular surface lacking smoothness. A similar granular, irregular surface was obtained for *Compsagis lesnei* Chopard, 1952, a smaller species (total length of 14.1 mm) with a punctuated pronotum.

This visual impression was confirmed by comparing the mesh distances calculated in MeshLab between pairs of models (Figs 5 and S1; S1 Table). For the 13 species, the average mesh distance ranged from 0.006 mm to 0.352 mm (with an across species average of 0.124 mm). When considering only the six least challenging species, this across species average dropped to 0,057 mm, which is in line with values obtained in another study for 19 bat skulls (0.054 mm) [52]. The lowest distances (e.g., 0.006 mm and 0.012 mm in *Cryptocercus parvus* and *Alloblatta nugax*, respectively) suggest that the 3D models generated through photogrammetry are of high quality, especially for non-challenging species.

To further assess the quality of the photogrammetric models and their usefulness for comparative morphometrics, we performed a PCA (Fig 6) where each species is plotted twice, one for each acquisition technique. The first two axes represented 80% of the variability. For most species, the pairs of models were very close to each other, although a few appeared to be further apart. Unsurprisingly, the latter corresponded to challenging species for photogrammetry (e.g., *Phortioeca peruana*, *Compsagis lesnei*, *Cryptocercus punctulatus* Scudder, 1862). Note that for species like *Hemelytroblatta africana*, bristles are not considered during µCt-scan acquisition, which largely contributes to the difference between the two acquisition techniques. Nonetheless, the technique of acquisition was found non-significant with this set of 13 species that included several photogrammetric challenging species (F = 1.1444; p-value MANOVA = 0.6408). This suggests that the differences induced by the use of two methods to reconstruct 3D models are negligible, and the variability

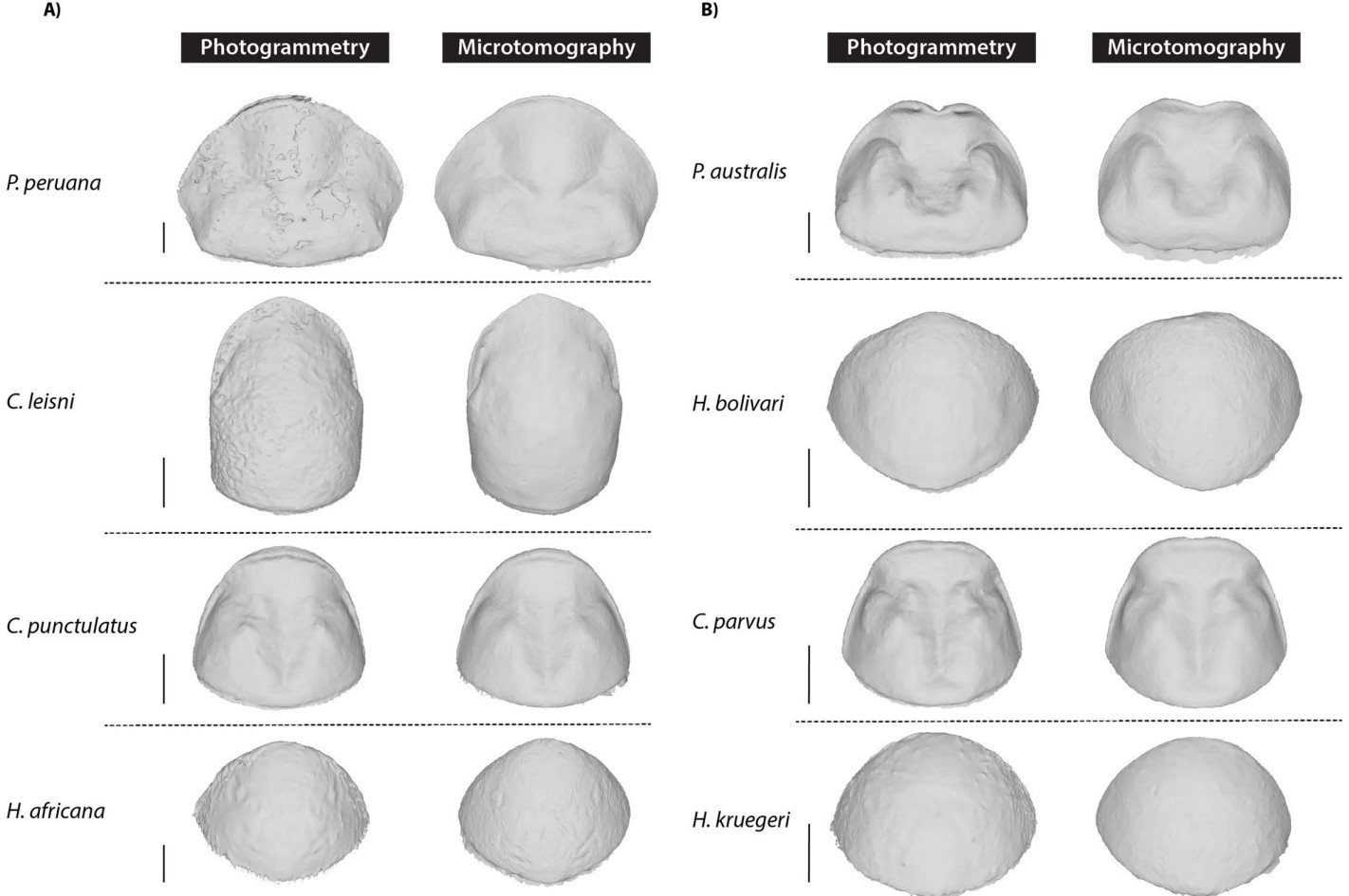

**Fig 4. Pronotums of eight species digitized by photogrammetry and microtomography (μCt-scan).** A) Four challenging species for photogramme-try: *Phortioeca peruana* (Blaberidae), *Compsagis leisni* (Blaberidae), *Cryptocercus punctulatus* (Cryptocercidae) and *Hemelytroblatta africana* (Corydii-dae). **B)** Four species with 3D models of comparable quality using both methods: *Panesthia australis* (Blaberidae), *Heterogamisca bolivari* (Corydiidae), *Cryptocercus parvus* (Cryptocercidae), *Heterogamisca kruegeri* (Corydiidae). Scale bars: 2 mm.

observed between specimens has biological causes and is not due to a methodological bias. We conclude that, especially for non-challenging species, 3D models obtained by our photogrammetric pipeline are of good enough quality to be used in comparative morphometric analyses.

## Limitations: A heterogeneous model quality

The main advantage of this method, its speed, has its drawback: the quality of the model is heterogeneous. By focusing on a specific area of the specimen for downstream geometric morphometrics analyses [33], the quality is here maximized on the pronotum. Quality deteriorates further away from the pronotum, which is particularly problematic for delicate struc-tures like bristles, antennae and legs. While the overall 3D models produced are satisfactory for dissemination purposes or for studying conspicuous morphological features, they would probably not be of sufficient quality for geometric morpho-metric analyses outside the region of interest.

   

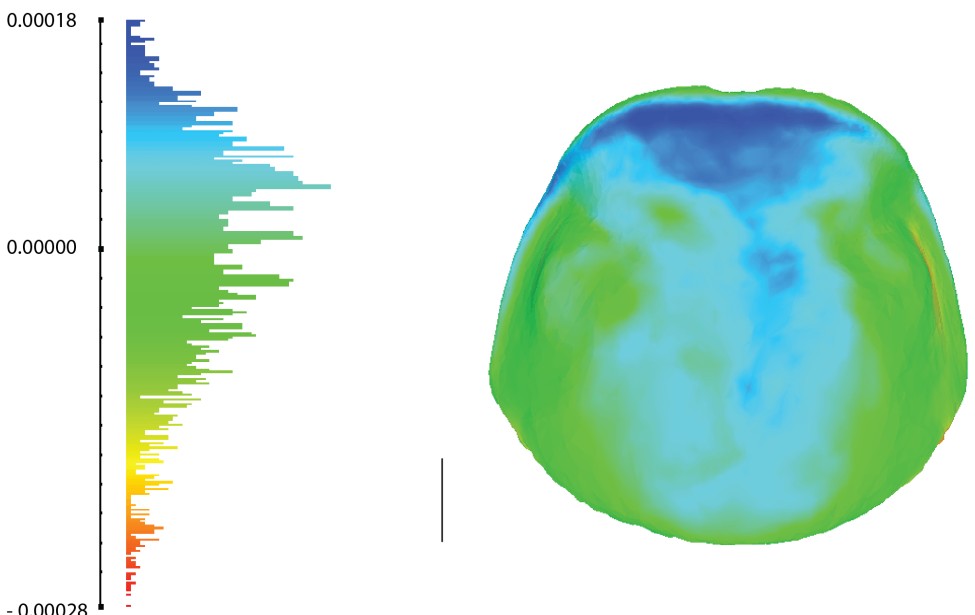

**Fig 5. Histograms (left) of the calculated distances between a target mesh (photogrammetric model) and a reference mesh (µCT-scan model).** These color-coded distances were used to color the pronotum (right), here of *Cryptocercus parvus*. The scale on the left shows that, even in the very limited region shown in warm colors, the difference between the two models is extremely small. Scale bar = 1 mm.

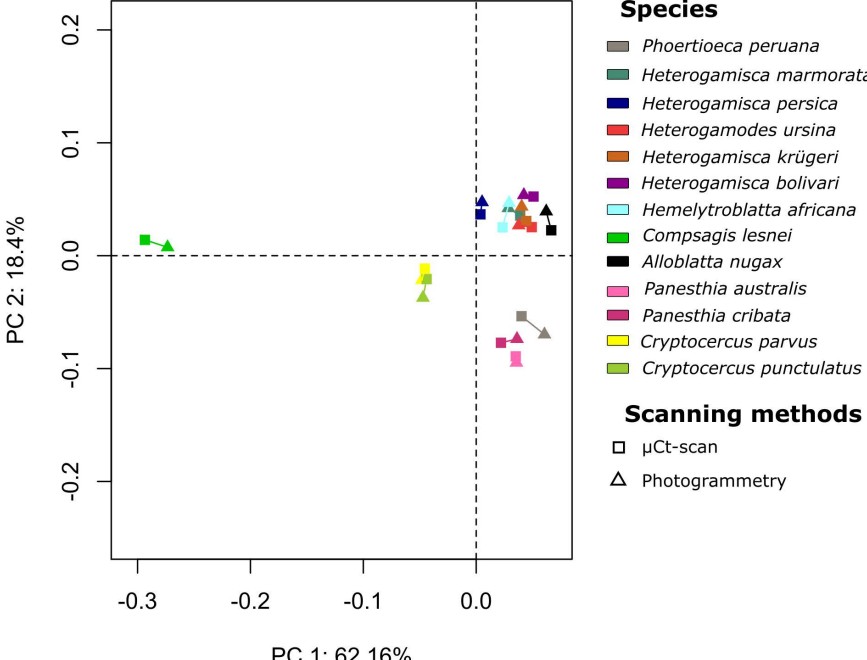

**Fig 6. PCA computed to compare the two digitization techniques, µCt-scan and photogrammetry.** For each species, the distance between the two models is small, in most cases smaller than the interspecific distances, suggesting a negligible effect of the acquisition method.

In addition, our pipeline does not solve common problems in photogrammetry. For instance, modeling wings by photogrammetry can be complex, when they are transparent or weakly contrasted, while reflective or deep-black surfaces can cause problems by interfering with the detection of key points by photogrammetric algorithms [57]. Likewise, hard-to-reach areas can be poorly modeled if the camera cannot directly capture these areas, which might then require additional acquisitions [13]. Our pipeline was not meant to address these issues.

## Conclusion

3D models have opened up a wide range of potential analyses and applications, among which the possibility to conduct geometric morphometric analyses. But, regarding photogrammetry, the examples are much more common for organisms bigger than insects [e.g., 44,58–61]. The main reason for this is probably that, for small objects, it takes a long time to obtain several decent models because of the need for focus stacking. We developed a methodology allowing to generate 3D models fast enough to gather a comparative dataset useful for ecomorphological study of insects [33], which was so far mainly accessible for other organisms or structures [e.g., 52,62,63].

By comparing photogrammetric and microtomographic 3D models, we showed that our inexpensive and time-saving pipeline not only produces models in large quantity but also in a quality good enough to perform geometric morphometric analyses. The negligible difference between 3D models generated by both techniques suggests that dataset combining models from these two acquisition modes can be used in morphometric analyses [40]. Given the interest in these analyses—consider for instance the *ca.* 2,000 citations of the package geomorph [54]—we hope that our pipeline will generate opportunities for the study of small objects like insects, one of the most species-rich group on Earth [64]. Finally, because 3D photogrammetric models are textured, making them available could contribute to various dissemination purposes and allow taxonomists to access specimens remotely.

## Supporting information

**S1 Fig. Color-coded representation of the distances calculated between a target mesh (photogrammetric model) and a reference mesh (µCT-scan model) for the pronotums of 13 cockroach species.** SD = standard deviation; scale bars = 1 mm.
(TIF)

**S1 Table. Distances (in millimeters) between a target mesh (photogrammetric model) and a reference mesh (µCT-scan model) computed for each pair of models.** Species are ordered according to the absolute value of average distance between meshes.
(DOCX)

**S2 Table. Summary of 3D reconstruction parameters obtained using two methods: photogrammetry and micro-computed tomography (µCT-scan).** Together, these data summarize the imaging and processing parameters used to generate the 3D models analyzed in this study.
(XLSX)

## Acknowledgments

We acknowledge Renaud Lebrun of the MRI platform member of the national infrastructure France-BioImaging. We thank Sylvain Gerber and Romain Garrouste (ISYEB, MNHN) for fruitful discussions about our project and photogrammetry. We also thank Steven Ahoua for his internship under JB and FL supervision, during which the first photogrammetric tests were conducted.

## Author contributions

**Conceptualization:** Juliette Berger, Frédéric Legendre.

**Data curation:** Juliette Berger, Barbara Bignon.

**Formal analysis:** Juliette Berger, Barbara Bignon.

**Funding acquisition:** Frédéric Legendre.

**Investigation:** Arnaud Delapré.

**Methodology:** Juliette Berger, Barbara Bignon, Raphaël Cornette, Arnaud Delapré.

**Project administration:** Frédéric Legendre.

**Software:** Juliette Berger, Barbara Bignon, Raphaël Cornette, Arnaud Delapré.

**Supervision:** Juliette Berger, Raphaël Cornette, Frédéric Legendre, Arnaud Delapré.

**Validation:** Raphaël Cornette, Arnaud Delapré.

**Writing – original draft:** Juliette Berger, Barbara Bignon, Frédéric Legendre.

**Writing – review & editing:** Juliette Berger, Barbara Bignon, Raphaël Cornette, Frédéric Legendre, Arnaud Delapré.

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
