## [Decision Letter · Decision Letter 0]

17 Mar 2025

Dear Dr. Berger,

Thank you for submitting your manuscript to PLOS ONE. After careful consideration, we feel that it has merit but does not fully meet PLOS ONE’s publication criteria as it currently stands. Therefore, we invite you to submit a revised version of the manuscript that addresses the points raised during the review process.

**ACADEMIC EDITOR:**

**Dear Authors,**

After the evaluations provided by independent experts in the field, my recommendation is that you revise your manuscript to adequately address the considerations raised by the reviewers.

Additionally, I would like to suggest emphasizing the comparison between photogrammetric and microtomographic 3D models, as this could be a fundamental aspect of your study. Strengthening this discussion would add significant value to your work and enhance its relevance to the scientific community.

**Best regards,**

**The Associate Editor**

We look forward to receiving your revised manuscript.

Kind regards,

Jose Balsa-Barreiro

Academic Editor

PLOS ONE

Journal Requirements:

Additional Editor Comments:

Dear Authors,

After the evaluations provided by independent experts in the field, my recommendation is that you revise your manuscript to adequately address the considerations raised by the reviewers.

Additionally, I would like to suggest emphasizing the comparison between photogrammetric and microtomographic 3D models, as this could be a fundamental aspect of your study. Strengthening this discussion would add significant value to your work and enhance its relevance to the scientific community.

Best regards,

The Associate Editor

Reviewers' comments:

Reviewer's Responses to Questions

**Comments to the Author**

1. Is the manuscript technically sound, and do the data support the conclusions?

Reviewer #1: Partly

Reviewer #2: Yes

Reviewer #3: Yes

Reviewer #4: Yes

Reviewer #5: Yes

2. Has the statistical analysis been performed appropriately and rigorously?

Reviewer #1: No

Reviewer #2: Yes

Reviewer #3: Yes

Reviewer #4: Yes

Reviewer #5: Yes

3. Have the authors made all data underlying the findings in their manuscript fully available?

Reviewer #1: No

Reviewer #2: Yes

Reviewer #3: Yes

Reviewer #4: Yes

Reviewer #5: Yes

4. Is the manuscript presented in an intelligible fashion and written in standard English?

Reviewer #1: Yes

Reviewer #2: Yes

Reviewer #3: Yes

Reviewer #4: Yes

Reviewer #5: Yes

Reviewer #1: (1) It seems AF-S DX Micro NIKKOR 40 mm f/2.8G lens is auto focus lens with maximum focal length 40mm. In this case, focal length might have changed if authors used auto focus mode. Did authors fixed the focal length or used auto focus mode?

If focal length was fixed somehow, did authors conduct camera calibration? What are the interior orientation parameters (principal point offset, focal length, coefficient of radial lens distortion, coefficient or decentering lens distortion)? The camera calibration information should be described with associated condition equation.

If authors used auto focus mode for each scene, how did authors address these varying focal lengths problem? How did authors conduct camera calibration with difference focal lengths for each scene? Also, please describe why the authors didn't use fixed focal length lens in this case.

(2) The description about exterior orientation (EO) parameters results (position and attitude (rotation) of camera) are not sufficient. Please describe EO parameters results more in details.

(3) The precision (post adjustment statistics) and accuracy of triangulation results were not included in the manuscript. Please include them.

Reviewer #2: Overall comment:

This study provides valuable data on the use of photogrammetry as a tool for obtaining morphometrics from specific structures of insects, in this case cockroaches. The study's strength lies in providing an accessible, streamlined and feasible workflow that should improve research efforts in the field of entomology while reducing costs and optimizing processing time. However, some aspects of the Methods section require additional clarification for readers of PLoS One to fully appreciate the robustness of the approach. Additionally, several minor instances throughout the manuscript could use clarifications and rephrasings. Overall, I believe these points can be addressed by a thorough revision of the manuscript. Detailed comments and suggestions are provided below.

Specific comments:

ABSTRACT

L16-21: This contextualization is important and I find it lacking in the beginning of the Introduction. Consider briefly expanding your first paragraph to fully contextualize readers about the importance of such collections.

L21: " ...insect specimens, remain to be improved” – change to “remains”.

INTRODUCTION

L45: It does not limit, but warrants manipulation prevention. Please, consider rephrasing.

L53: I’m not sure what you mean by “keep them alive”. You mean the preservation of the NHC? Please, clarify and rephrase.

L56: The paper would gain from stating which other purposes Photogrammetry is also valuable for. For instance, you can cite many other advantages, like being able to maintain a digital register than can be revisited, being able to access precise metrics from different view points, among others.

L62: I do not understand “which can the thorny for type specimens”. Please, clarify.

L85: Consider improving the flow of the sentence when citing the cost of rapidity.

L89: Is this akin to a Macro photography? Consider better explaining how the focus on the region of interest eliminates the need for focus stacking. This is particularly important because that’s one of the main aspects of you study, and I feel more time should be invested in explaining how exactly this is achieved.

L92-93: Consider rephrasing to “...this pipeline can produce high-quality 3D models of the structure of interest that realistically capture...”

MATERIALS AND METHODS

L100: This sentence is na Objective and should be stated in the last paragraph of the Introduction, expanded in more details.

L101-103: Consider replacing the word “tool” for “structure” here and elsewhere. Also, the sentence would benefit by stating the focus on the body part in the beginning instead of the end.

L110: “Cockroaches”, in the plural.

L119: Usually we state numbers like these in the numeric form, i.e. 26 and 15. Small numbers, <10, are correctly stated throughout.

L120: I had trouble understanding what you were referring as challenging here. Only when I advanced in the text I understood. Please, consider clarifying this to make it easier for readers to understand exactly that the presence of bristles, reflection and black coloration are the challenges cited.

L125: The manuscript would benefit by keeping units consistent. Consider converting to mm.

L129: Here it is clearer which are the conditions that are challenging for photogrammetry, but you are talking about X-Rays? It is confusing; please, clarify.

L138: Replace “shots” for “images” of “pictures”.

L146-147: Only citing the portability of the Godox LST40 is irrelevant here. If portability is an important factor, as stated in the Results and Discussion, please clarify that the whole set-up is portable.

L168 Table 2: The price stated for Metashape here is different than the one in Results and Discussion (830 USD). Please, clarify.

L173: Brightness of the specimen? You mean brightness of the lighting conditions for each specimen? Please, clarify.

L208-221: This is an important step in generating high-quality models and I feel should be improved. It seems to be missing the step on integrating the masks generated in Photoshop to the workflow in Metashape. It would be important for readers to be able to fully reproduce the settings of this pipeline with full details.

L267: You mean 96 GB of RAM?

RESULTS AND DISCUSSION

L318: As I mentioned, there is a discrepancy in the values informed.

L319: This sentence in weirdly placed here. Consider rephrasing (e.g. the photographic set-up is portable, inexpensive and provides a fast workflow).

L324: Consider replacing “issues” for “limited”.

L327: Specify that you refer to “focus” stacking for consistency.

L367: Avoid superlatives such as 'ultra-high' when not referencing specific parameters from photogrammetry-based software. Consider stating that 3D models are of high quality, in a more conservative manner.

L376: Change to “PCA”.

L379: “Correspond” to “corresponded”.

L388: Consider rephrasing to “are of good enough quality”.

Reviewer #3: Berger and colleagues meticulously developed an optimal protocol for reproducing small objects like cockroaches. I personally enjoyed reading the paper and truly appreciated the attention to detail, the table on incurred costs, and all the effort and dedication involved in refining this approach without utilising photo stacking.

I have a question stemming from curiosity: although the primary focus was on the first dorsal thoracic segment (pronotum) and all attention was directed towards the dorsal features of the insects, I wondered whether additional elements, particularly on the ventral side, could also be included in the 3D model. Was this a limitation of the method, or do you believe that adjustments, such as positioning the insect higher on the pedestal (without that brown section of the equipment obscuring the ventral side, as illustrated in your Figure 2 Step 3), could be made to capture the ventral features as well?

Reviewer #4: The manuscript "Rapid, without focus stacking, 3D photogrammetric digitization of

cockroaches" presents an innovative methodology for digitizing insect specimens, specifically

cockroaches, for morphometric analysis. The authors propose a photogrammetric pipeline that

excludes the time-consuming process of focus stacking, which is traditionally used to increase

the depth of field in photogrammetry, particularly for small objects.

The authors present a well-structured study and apply a robust methodology to address the gap in the adoption of photogrammetry applications. Only minor details need to be addressed.

Including error values in the scale bars will further support the absence of significant biases between the two techniques used in your work.

The hyphen ("-") appears to be a typo or error (e.g., in lines 24 and 28). Correct this inconsistency throughout the document as needed.

Ensure consistency in word choice by selecting either "modeling" or "modelling" and applying it uniformly.

Reviewer #5: This study shows the advantages of a photogrammetric workflow to generate high quality 3D models of objects as small as insects (cockroaches). In general, I consider the work publishable after review and correction of some points:

The introduction fulfills its purpose by showing the reader the benefits of this workflow tested on cockroaches. However, I think that the introduction can be improved by including the following bibliography.

- Beth et al., (2012). Whole-drawer imaging for digital management and curation of a large entomological collection.

- Muthu et al. (2023). Towards end-to-end automatic insect handling and insect scanning.

- Thanh-Nghi Doan (2024). A low-cost digital 3D insect scanner.

- Mathys et al. (2024). Sphaeroptica: A tool for pseudo-3D visualization and 3D measurements on arthropods.

Line 125: Standardize the unit of measurement of the specimens, either in cm or mm. In line 125 they are in cm and in line 117 in mm.

Line 183: Mention what angles were used in previous or similar studies.

Line 231: Almost all photogrammetric processing with Agisoft Metashape software was done with high values, except for photo alignment. Why did you choose a medium value instead of a high value? There are other steps that are more time consuming than photo alignment (e.g. building a dense cloud). Also, increasing the alignment to "high" may improve the accuracy of your measurements.

Line 254: How many markers were placed on the photos? Were the measurements obtained from this scale assignment to the 3D models subsequently validated by measuring the specimens?

Line 267: Correct 96 Go by 96 GB

Line 311: Compare the results with the same or similar studies using photogrammetry.

**Do you want your identity to be public for this peer review?** For information about this choice, including consent withdrawal, please see our Privacy Policy

Reviewer #1: No

Reviewer #2: No

Reviewer #3: No

Reviewer #4: No

Reviewer #5: No

---

## [Author Response · Author response to Decision Letter 1]

9 Sep 2025

Dear PLoS One editorial board members, and reviewers,

We thank you for giving us the opportunity to revise and resubmit our study. We also thank the five reviewers, who provided constructive comments to improve our manuscript. We have carefully considered all comments and suggestions. Among the most important changes, we have now revised the manuscript by addressing the following main points raised by the reviewers:

1) We have improved the contextualization of this work and expanded on how it compares with other works (reviewers #2 and #5).

2) We have clarified several aspects of the method used (reviewers #1, #2 and #5).

---

## [Decision Letter · Decision Letter 1]

2 Nov 2025

Rapid, without focus stacking, 3D photogrammetric digitization of cockroaches

PONE-D-24-54510R1

Dear Dr. Berger,

We’re pleased to inform you that your manuscript has been judged scientifically suitable for publication and will be formally accepted for publication once it meets all outstanding technical requirements.

Kind regards,

Wesley D. Colombo

Academic Editor

PLOS ONE

Additional Editor Comments (optional):

Reviewers' comments:

Reviewer's Responses to Questions

**Comments to the Author**

Reviewer #2: All comments have been addressed

2. Is the manuscript technically sound, and do the data support the conclusions?

Reviewer #2: Yes

3. Has the statistical analysis been performed appropriately and rigorously?

Reviewer #2: Yes

4. Have the authors made all data underlying the findings in their manuscript fully available?

Reviewer #2: Yes

5. Is the manuscript presented in an intelligible fashion and written in standard English?

Reviewer #2: Yes

Reviewer #2: Thank you for carefully addressing the reviewers’ comments in your revision. The manuscript has been substantially improved in clarity and quality. I find that the concerns previously raised have been adequately resolved, and the paper now meets the standards for publication. I have no further substantive comments.

**Do you want your identity to be public for this peer review?** For information about this choice, including consent withdrawal, please see our Privacy Policy

Reviewer #2: No

---

## [Editor Report · Acceptance letter]

PONE-D-24-54510R1

PLOS ONE

Dear Dr. Berger,

I'm pleased to inform you that your manuscript has been deemed suitable for publication in PLOS ONE. Congratulations! Your manuscript is now being handed over to our production team.

Kind regards,

on behalf of

Dr. Wesley D. Colombo

Academic Editor

PLOS ONE